# The Prognostic Biomarkers of Plasma Trimethylamine N-Oxide and Short-Chain Fatty Acids for Recanalization Therapy in Acute Ischemic Stroke

**DOI:** 10.3390/ijms241310796

**Published:** 2023-06-28

**Authors:** Ping-Song Chou, I-Hsiao Yang, Chia-Ming Kuo, Meng-Ni Wu, Tzu-Chao Lin, Yi-On Fong, Chi-Hung Juan, Chiou-Lian Lai

**Affiliations:** 1Department of Neurology, Kaohsiung Medical University Hospital, Kaohsiung Medical University, Kaohsiung 807378, Taiwan; pschou1013@gmail.com (P.-S.C.); berkeley0701@gmail.com (M.-N.W.); prochristy@gmail.com (T.-C.L.); amyfong0813@gmail.com (Y.-O.F.); 2Department of Neurology, Faculty of Medicine, College of Medicine, Kaohsiung Medical University, Kaohsiung 807377, Taiwan; 3Graduate Institute of Medicine, College of Medicine, Kaohsiung Medical University, Kaohsiung 807377, Taiwan; 4Neuroscience Research Center, Kaohsiung Medical University, Kaohsiung 807377, Taiwan; 5Department of Medical Imaging, Kaohsiung Medical University Hospital, Kaohsiung Medical University, Kaohsiung 807378, Taiwan; toni_yihsiao@yahoo.com.tw; 6Department of Nursing, Kaohsiung Medical University Hospital, Kaohsiung Medical University, Kaohsiung 807378, Taiwan; 960162@mail.kmuh.org.tw; 7Institute of Cognitive Neuroscience, National Central University, Taoyuan City 320317, Taiwan; chijuan@cc.ncu.edu.tw; 8Cognitive Intelligence and Precision Healthcare Research Center, National Central University, Taoyuan City 320317, Taiwan

**Keywords:** acute ischemic stroke, endovascular thrombectomy, intravenous thrombolysis, isovalerate, short-chain fatty acids, trimethylamine N-oxide

## Abstract

Bidirectional communication of the microbiota–gut–brain axis is crucial in stroke. Recanalization therapy, namely intravenous thrombolysis (IVT) and endovascular thrombectomy (EVT), are recommended for eligible patients with acute ischemic stroke (AIS). It remains unclear whether gut microbiota metabolites, namely trimethylamine N-oxide (TMAO) and short-chain fatty acids (SCFAs), can predict the prognosis after recanalization therapy. This prospective study recruited patients with AIS receiving IVT, EVT, or both. The National Institutes of Health Stroke Scale (NIHSS) and modified Rankin scale (mRS) scores were used to assess the severity and functional outcomes of AIS, respectively. A functional outcome of mild-to-moderate disability was defined as a mRS score of 0–3 at discharge. Plasma TMAO and SCFA levels were measured through liquid chromatography with triple-quadrupole mass spectrometry. Fifty-six adults undergoing recanalization therapy for AIS were enrolled. Results showed that TMAO levels were not associated with stroke severity and functional outcomes, while isovalerate levels (one of the SCFAs) were negatively correlated with NIHSS scores at admission and discharge. In addition, high isovalerate levels were independently associated with a decreased likelihood of severe disability. The study concluded that an elevated plasma isovalerate level was correlated with mild stroke severity and disability after recanalization therapy for AIS.

## 1. Introduction

Stroke is the second leading cause of death worldwide and causes disability in a large proportion of survivors [1], with ischemic stroke accounting for approximately 80% of strokes [2]. Numerous studies have identified risk factors for stroke and developed therapeutic strategies to reduce the burden of stroke. Recanalization therapy, termed “intravenous thrombolysis” (IVT) and “endovascular thrombectomy” (EVT), is recommended for eligible patients with acute ischemic stroke (AIS) [3]. Factors that predict the outcomes of IVT and EVT in patients with stroke have been proposed on the basis of clinical characteristics, neuroimaging findings, and initial stroke severity. However, whether microbiological factors can predict the outcomes of recanalization therapy following AIS remains unclear.

“Microbiota” refers to the collection of microorganisms, including bacteria, archaea, viruses, and single-celled eukaryotes, which live in coexistence with the human body, colonizing mainly the gastrointestinal tract. The gut microbiota can produce bioactive metabolites and communicate with host organs through complex pathways that can support or affect host physiological functions [4,5]. The gut microbiota might play a crucial role in human health and diseases and is thus considered a new potential therapeutic target [4].

Bidirectional communication, referred to as the “microbiota–gut–brain axis”, plays a crucial role in the relationship between the gut microbiota and stroke [6]. Dysbiosis, an imbalance in the composition and function of the gut microbiota, is associated with numerous risk factors for stroke, including atherosclerosis, coronary heart disease, hypertension, and type 2 diabetes mellitus [5]. Major ischemic stroke induces dysbiosis, which in turn affects stroke outcomes by causing neuroinflammation [7,8]. In addition, alterations in gut microbiota appear to be connected with bowel function, daily activities, and motor function of poststroke patients [9].

In addition to the composition of the gut microbiota, its metabolites, namely trimethylamine N-oxide (TMAO) and short-chain fatty acids (SCFAs), contribute to the development and outcomes of stroke. TMAO, a gut microbiota metabolite transferred from dietary choline and carnitine, has been reported to be associated with increased atherogenesis and thrombus formation [10]. An increasing amount of evidence indicates that TMAO is both a risk factor for and a prognostic predictor of stroke [11,12]. In addition, an elevated level of TMAO was associated with an increased risk of recurrent stroke in patients with cerebral small vessel disease [13], independent of dual antiplatelet and lipid-lowering therapy [14].

SCFAs result from the fermentation of dietary fiber by the gut microbiota. SCFAs reduce cardiovascular disease risk through glucose homeostasis, immunomodulation, appetite regulation, and obesity [15]. However, the effect of SCFAs on ischemic stroke is controversial and varies by the type of SCFA. Deficiencies of SCFA-producing bacteria and fecal SCFAs were observed in patients with AIS and associated with severe stroke and poor functional outcomes [16]. Among SCFAs, acetate, valeric acid, and butyrate were negatively correlated with neurological deficits and infarct volume in an animal model of middle cerebral artery occlusion [17]. A low level of acetate was reported to be associated with an increased risk of unfavorable functional outcomes in patients with AIS [16]. In contrast, isobutyrate, butyrate, and 2-methylbutyrate were demonstrated to be associated with poor symptom recovery and severe neurological deficits in patients with AIS receiving EVT [18].

According to the literature review above, the effects of different SCFAs on the severity and outcomes of AIS remain to be determined. In addition, the role of TMAO and SCFAs in predicting outcomes following recanalization therapy for AIS is not well established. This study investigated clinical and molecular associations between circulating TMAO and SCFAs and AIS after recanalization therapy.

## 2. Results

### 2.1. Patient Demographics

Fifty-six adults underwent recanalization therapy for AIS. The mean age of the patients was 70.7 (±13.0) years, and 32 (57.1%) patients were male. Hypertension was present in 46 (82.1%) patients and was the most common vascular risk factor. A total of 42 (75.0%) and 27 (48.2%) patients had hyperlipidemia and atrial fibrillation/flutter, respectively. Patients were discharged a median of 19 (interquartile range, 12–39) days after admission.

Upon admission, the median NIHSS score was 15 (interquartile range, 10.3–21). Forty-three (76.8%) patients experienced neurological improvement after recanalization therapy. At discharge, the median NIHSS score was 7 (interquartile range, 3–15). Twenty-three (41.1%) patients had outcomes of mild-to-moderate disability. The patients with mild-to-moderate disability had a significantly lower prevalence of hypertension and atrial fibrillation/flutter than those with severe disability. The patients with mild-to-moderate disability had significantly lower initial and discharge NIHSS than those with severe disability. Table 1 summarizes the comparison between demographic data, the prevalence of vascular risk factors, and stroke severity between groups.

### 2.2. Details of Recanalization Therapy

Out of the total number of patients, 30 (53.6%) patients received IVT, 15 (26.8%) received EVT, and 11 (19.6%) received both IVT and EVT. The mild-to-moderate disability group had a significantly higher rate of IVT alone than the severe disability group. The median time from stroke onset to IVT was 110 (interquartile range, 95–199) and 121 (interquartile range, 95.8–153.3) minutes in the mild-to-moderate and severe disability groups, respectively. The median time from stroke onset to reperfusion in the patients receiving EVT was 280 (interquartile range, 202–420.5), and 388 (interquartile range, 310–453.5) minutes in the mild-to-moderate and severe disability groups, respectively. Stroke onset to IVT, groin puncture, and reperfusion time did not significantly differ between the groups. The percentage of the patients with successful reperfusion (mTICI score ≥ 2b) after EVT was 100% and 85.7% in the mild-to-moderate and severe disability groups, respectively. The majority (80.4%) of occluded vessels were in the anterior circulation, with similar proportions in each group. Appendix A summarizes the details of recanalization therapy.

### 2.3. Plasma TMAO Levels and Stroke Outcomes

Differences in the overnight fasting level of TMAO after admission were evaluated in relation to patient demographics, and no significant differences in the TMAO level based on sex and vascular risk factors were noted (Appendix A). No significant correlation was observed between the TMAO level and age, body mass index (BMI), and lipid profiles. A significant positive correlation was noted between the TMAO level and glycated hemoglobin (HbA1c; Pearson correlation coefficient 0.400, *p* = 0.002; Appendix A). Spearman rank correlation results revealed a trend toward a positive correlation between the TMAO level and NIHSS scores upon admission and at discharge (Table 2; Figure 1). The TMAO level did not differ significantly between the two outcome groups (Table 3; Figure 2).

**Table 2 ijms-24-10796-t002:** Spearman rank correlation between the NIHSS score and plasma TMAO and SCFA levels.

Spearman Rank Correlation	All Enrolled Patients with Stroke (n = 56)			
	Initial NIHSS	*p* Value	Discharge NIHSS	*p* Value
TMAO	0.087	0.524	0.230	0.089
Formate	0.089	0.516	−0.028	0.836
Acetate	0.077	0.578	−0.003	0.983
Propionate	0.019	0.891	0.078	0.566
Isobutyrate	0.182	0.179	0.099	0.468
Butyrate	0.090	0.511	0.165	0.225
2-methylbutyrate	0.250	0.063	0.209	0.122
Isovalerate	−0.329	0.013 *	−0.370	0.005 **
Valerate	0.168	0.217	0.185	0.173

Abbreviations: NIHSS, National Institute of Health Stroke Scale; SCFA, short-chain fatty acids; TMAO, trimethylamine N-oxide. * indicated *p* < 0.05. ** indicated *p* < 0.01.

**Figure 1 ijms-24-10796-f001:**
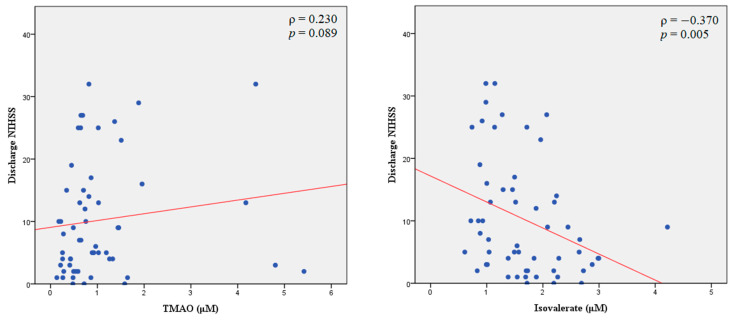
Spearman rank correlation coefficient (***ρ***) and *p*-value (*p*) between the plasma levels of trimethylamine N-oxide (TMAO) and isovalerate, with the NIHSS score at discharge in patients with acute ischemic stroke. The red line is the line of best fit. NIHSS, National Institute of Health Stroke Scale.

**Table 3 ijms-24-10796-t003:** Differences in plasma TMAO and SCFA levels between patients with mild-to-moderate disability and severe disability.

Characteristic	Total	Mild-to-Moderate Disability (n = 23)	Severe Disability (n = 33)	*p* Value
TMAO, μM, mean (±SD)	1.1 ± 1.1	1.1 ± 1.4	1.1 ± 0.9	0.983
Formate, μM, mean (±SD)	75.2 ± 17.0	76.8 ± 18.5	74.1 ± 16.0	0.564
Acetate, μM, mean (±SD)	136.0 ± 58.8	137.0 ± 59.4	135.3 ± 59.3	0.916
Propionate, μM, mean (±SD)	10.1 ± 3.4	10.3 ± 3.7	9.9 ± 3.2	0.688
Isobutyrate, μM, mean (±SD)	17.4 ± 6.1	17.4 ± 5.9	17.4 ± 6.3	0.972
Butyrate, μM, mean (±SD)	4.1 ± 5.8	5.1 ± 9.0	3.5 ± 1.3	0.310
2-methylbutyrate, μM, mean (±SD)	1.2 ± 0.4	1.2 ± 0.4	1.2 ± 0.5	0.797
Isovalerate, μM, mean (±SD)	1.7 ± 0.7	2.0 ± 0.6	1.5 ± 0.7	0.013 *
Valerate, μM, mean (±SD)	1.0 ± 0.4	0.9 ± 0.4	1.0 ± 0.5	0.765

Abbreviations: SCFA, short-chain fatty acids; SD, standard deviation; TMAO, trimethylamine N-oxide. * indicated *p* < 0.05.

**Figure 2 ijms-24-10796-f002:**
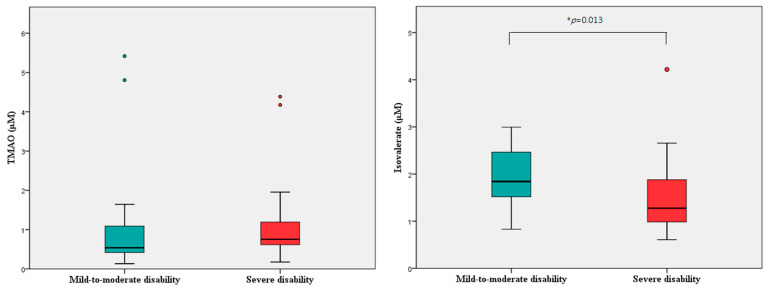
Box–Whisker plots indicating the plasma levels of trimethylamine N-oxide (TMAO) and isovalerate, in patients with acute ischemic stroke with mild-to-moderate disability and severe disability. Green color indicated mild-to-moderate disability group; red color indicated severe disability group. * indicated *p* < 0.05.

### 2.4. Plasma SCFA Levels and Stroke Outcomes

When the SCFA level was compared by sex, no significant difference was noted between the male and female. Significant decreases were observed in the formate level in the patients with hypertension (73.0 ± 14.5 vs. 85.5 ± 23.8, *p* = 0.033), the isobutyrate level in the patients with hyperlipidemia (16.2 ± 5.3 vs. 20.9 ± 7.1, *p* = 0.012), and the isovalerate level in the patients with atrial fibrillation/flutter (1.4 ± 0.5 vs. 1.9 ± 0.8, *p* = 0.014) compared with those without these vascular risk factors (Appendix A). The acetate level was significantly and negatively correlated with BMI (Pearson correlation coefficient = −0.319, *p* = 0.020). In the biochemical examination, the acetate level was negatively correlated with the total cholesterol level (Pearson correlation coefficient = −0.289, *p* = 0.032) and the LDL level (Pearson correlation coefficient = −0.275, *p* = 0.042; Appendix A).

The isovalerate level was negatively correlated with the NIHSS score upon admission (Spearman’s rank correlation coefficient = −0.329, *p* = 0.013) and at discharge (Spearman’s rank correlation coefficient = −0.370, *p* = 0.005), but other SCFAs were not associated with NIHSS scores (Table 2; Figure 1). Isovalerate was the only SCFA that was significantly associated with outcomes of AIS, and the isovalerate level was higher in the patients with mild-to-moderate disability than in those with severe disability (2.0 ± 0.6 vs. 1.5 ± 0.7 μM, *p* = 0.013; Table 3; Figure 2). Furthermore, Table 4 presents the results of the multivariate logistic regression analysis, which examines the relationship between isovalerate level and the outcome of severe disability. The isovalerate level was found to be associated with a decreased likelihood of severe disability independent of covariates in demographic data [odds ratio (OR) = 0.32; 95% confidence interval (CI) = 0.13–0.79, *p* = 0.013], comorbid vascular risk factors (OR = 0.27; 95% CI = 0.09–0.79, *p* = 0.017), and the treatment of EVT (OR = 0.32; 95% CI = 0.11–0.95, *p* = 0.041). These multivariate logistic models were well calibrated as assessed by the Hosmer and Lemeshow goodness-of-fit test.

### 2.5. Subgroup Analysis of the Effect of Isovalerate on Functional Outcomes

Using the median isovalerate level of 1.54 μM as the cutoff, the effect of isovalerate on functional outcomes was examined in several patient subgroups (Figure 3). Among the subgroups, a significant interaction was observed only between age and isovalerate level. An increased isovalerate level was associated with a higher likelihood of mild-to-moderate disability in patients aged < 75 years (*p* for interaction = 0.017). No significant interaction was found between the isovalerate level and other covariates.

## 3. Discussion

To the best of our knowledge, no study has determined the prognostic value of TMAO and SCFAs for functional outcomes after recanalization therapy in patients with AIS. This study investigated the role of TMAO and SCFAs in relation to stroke severity and functional outcomes after recanalization therapy. The results revealed that isovalerate, a branched SCFA (BCFA), was associated with mild stroke severity and disability after recanalization therapy for AIS, particularly in patients aged younger than 75 years.

Although the TMAO level could predict stroke severity [19] and unfavorable functional outcomes after AIS [20], the results of this study do not indicate an association between the TMAO level and stroke severity and functional outcomes after recanalization therapy for AIS. This finding can be explained by the low plasma TMAO levels and high NIHSS scores of our enrolled patients. The plasma TMAO cutoff used to predict unfavorable outcomes after stroke has been reported to range from approximately 4.95 to 6.6 μM [19,21]. However, in our patients, the median plasma TMAO level was 0.71 (interquartile range, 0.45–1.24) μM, and the maximum value was 5.42 μM. In addition, because the enrolled patients were indicated for recanalization therapy, they had a higher initial NIHSS score (median = 15) than those in other studies (median = approximately 4 to 10) [19,20,21]. In summary, in the population with initial severe AIS and low TMAO levels, TMAO cannot be used to predict functional outcomes after recanalization therapy for AIS.

The TMAO levels in our study were measured within 24 h after the onset of AIS, which has been shown to be a reliable measurement in previous studies that demonstrated no significant changes in TMAO levels during the acute stroke period [22,23]. However, our recruited patients had relatively low TMAO levels. In addition to the different dietary habits and lifestyles, the explanation for the discrepancy in TMAO levels between this study and previous studies could be due to the treatment for hyperlipidemia. Statin treatment has been reported to lower TMAO levels [24], and 75% of our study patients had comorbid hyperlipidemia, defined by either receiving or having received statin treatment in the past. Hence, statin treatment may contribute to the lower TMAO levels in our study population.

SCFAs, bacterial metabolites produced by the fermentation of dietary fibers, not only maintain the intestinal immune system but also modulate communication in the microbiota–gut–brain axis in stroke [25]. Although changes in SCFA-producing bacteria and SCFA levels were observed in patients with AIS, some discrepancies were observed among the types of SCFAs. Increased SCFA-producing bacteria (including *Odoribacter* and *Akkermansia*) were noted in patients with AIS; however, their correlation with stroke severity and outcomes differed among bacteria [26]. A decreased abundance of *Fecalibacterium*, a butyrate producer, was observed in patients with AIS [8]. AIS is associated with decreased fecal acetate levels and increased valerate levels, which are related to host metabolism and inflammation [27]. Loss of SCFA-producing bacteria and low fecal acetate levels increased the risk of poor functional outcomes in patients with AIS [16,28]. In contrast, high plasma levels of isobutyrate, butyrate, and 2-methylbutyrate were reported to be associated with poor neurological recovery and severe neurological deficits after EVT for AIS [18]. These discrepancies may be due to differences in the background of enrolled participants, the initial severity of the stroke, the sampling of stool or blood, and the timing of sampling from the onset of stroke symptoms.

Isovalerate represents a small fraction of the BCFA pool, and the effect of BCFAs on stroke remains largely unknown. This study demonstrated that an increased plasma isovalerate level after recanalization therapy was associated with mild neurological deficits and mild disabling outcomes after AIS. Following a multivariate logistic regression analysis, it was found that there was a significant association between the isovalerate level and decreased disability after recanalization therapy in AIS patients, which was independent of patients’ demographic data and comorbid vascular risk factors. However, when analyzing the effect of EVT (EVT only and combined IVT and EVT) on outcomes and including it in the model, there was a significant and independent association between EVT and severe disability after recanalization therapy. The reason may be that the treatment of EVT was not randomized in this study and depended on the patient’s initial severity of AIS. EVT was only indicated for patients with high severity (NIHSS ≥ 8) and major vessel occlusion according to the guideline of AIS, so the association between EVT and the increased risk of severe disability may be due to the high initial severity of such patients (median initial NIHSS, IVT only: 12, EVT: 21). Nevertheless, the association between isovalerate and decreased disability persisted even when accounting for all variables. Subgroup analysis revealed no interaction between recanalization therapy and isovalerate levels in terms of treatment outcomes. Taken together, this indicated that isovalerate was associated with decreased disability after recanalization therapy for AIS. Reperfusion status was not included in the regression analysis as it was only assessed for patients who underwent EVT, not for those receiving IVT only.

The results of the subgroup analysis showed that there was a significant interaction effect between isovalerate and age on the outcome. Specifically, in patients aged < 75 years, isovalerate was significantly associated with decreased disability, but this association was not observed in patients aged ≥ 75 years. This difference may be explained by the fact that age is an important predictor of poor prognosis in patients receiving IVT and EVT [29,30]. In this study, patients aged ≥ 75 years had a higher prevalence of two or more comorbid vascular risk factors than those aged < 75 years (79.2% vs. 40.0%), as well as a lower reperfusion rate after EVT (TICI > 2b, 62.5% vs. 100%), which could result in a significantly lower NIHSS improvement during hospitalization (−1.71 vs. −8.41). Therefore, in older patients, the association of isovalerate with the outcome may be diminished.

In an animal model, significantly decreased fecal levels of SCFAs, including isovalerate, were observed in aged mice, and the reduction was associated with increased inflammation and poor outcomes after middle cerebral artery occlusion [31]. Aged stroke mice receiving SCFA producers and inulin exhibited increased gut, brain, and plasma levels of isovalerate and the amelioration of poststroke neurological deficits and inflammation [32]. Changes in fecal total SCFA levels, including isovalerate, after treatment with tanshinone IIA–loaded nanoparticles and neural stem cell combination therapy in the acute stage were negatively correlated with ischemic lesion volume and midline shift in pigs with middle cerebral artery occlusion [33]. These results suggest that isovalerate is involved in the communication between the gut and brain and can improve functional recovery after recanalization therapy in patients with AIS. However, given the lack of studies investigating the effect of changes in BCFA on stroke outcomes, additional studies exploring the effects of the composition of the gut microbiota on the isovalerate level and outcomes after recanalization therapy for AIS are warranted.

This is the first study to examine the plasma levels of TMAO and SCFAs in patients with AIS undergoing recanalization therapy and demonstrate the association between plasma isovalerate levels and AIS outcomes. However, this study has several limitations that should be considered to interpret the results appropriately. First, dynamic changes in the levels of TMAO and SCFAs happened after AIS. Thus, the timeframe of blood sampling may affect the final results. No significant changes were observed in the TMAO level before and within 24 h of AIS, but the level decreased significantly thereafter [22,23]. Therefore, the fasting blood sample collected within 24 h of AIS in the study was used to determine the baseline TMAO level of the patients enrolled. However, evidence of dynamic changes in the plasma SCFA level after stroke is lacking. Most studies have used the fecal SCFA level, whereas one study used the plasma SCFA level during EVT [18]. The appropriate time point for blood sampling to determine the effect of SCFAs on AIS remains unknown, and the quantification of the plasma SCFA level before, at the time of, and after AIS in a study of the same patients would be ideal. Second, we did not determine whether the plasma levels of TMAO and SCFAs correspond to their brain levels. Research has reported that after gavage with SCFA producers with inulin, both the plasma and brain levels of SCFAs increased in mice [32], suggesting a correlation between the plasma and brain levels of SCFAs. However, because of the lack of evidence indicating the correlation between the plasma and brain levels of TMAO and the difficulty in determining the brain level of TMAO in humans, studies have mostly determined the association of TMAO with stroke outcomes using the plasma levels. Therefore, using the plasma levels of TMAO and SCFAs can be acceptable in this study.

Third, this study included patients with AIS who received IVT, EVT, or a combination of both. As IVT and EVT have time restrictions, only a small fraction of patients are eligible for these treatments. Hence, due to the limited number of participants, IVT and EVT were not separately analyzed in this study. Additionally, since IVT and EVT have different definitions of favorable functional outcomes in previous studies, this study used the unified cut-off mRS score of 3 to indicate mild-to-moderate disability after recanalization therapy. Finally, the study had a short follow-up period, with functional outcomes based on the mRS score at discharge, mostly within 30 days. Previous studies investigated the effects of TMAO and SCFAs on long-term functional outcomes mainly using the 90-day mRS score. This study focused on the early effect of TMAO and SCFAs on AIS outcomes after recanalization therapy. Future studies should control and address these confounding factors to determine the association between the gut microbiota, plasma isovalerate level, and the clinical consequences of AIS after recanalization therapy.

## 4. Materials and Methods

### 4.1. Patients

This single-center prospective cohort study was conducted in southern Taiwan. Patients were enrolled if they (1) had AIS and were admitted within 6 h of symptom onset; (2) were treated with recanalization therapy, including IVT and/or EVT and (3) received brain magnetic resonance imaging (MRI) after admission and acute ischemic lesions were detected on brain MRI. Patients were excluded if they were aged < 20 years, had radiological evidence of an initial intracerebral hemorrhage, were ineligible for IVT or EVT, or received probiotics or antibiotics within 1 week prior to admission. A comprehensive assessment of demographic data, medical history, physical and neurological status, and blood biochemistry was performed for each patient.

### 4.2. Stroke Severity, Functional Outcomes, and Reperfusion Assessment

Stroke severity was assessed by determining the National Institutes of Health Stroke Scale (NIHSS) score, and functional outcomes were evaluated by determining the modified Rankin Scale (mRS) score. The NIHSS score ranges from 0 to 42 and indicates the level of neurological impairment. Higher scores indicate severe neurological deficits. The NIHSS score was determined before treatment and at discharge. The mRS score was used to assess functional status at baseline and at discharge. A functional outcome of mild-to-moderate disability was defined as an mRS score of 0–3 at discharge. The NIHSS and mRS scores were measured by a neurologist masked to patients’ TMAO and SCFA levels. Reperfusion status after EVT was examined using the modified treatment in cerebral ischemia (mTICI) score, and the mTICI score was determined by the EVT operator by using the final angiograms.

### 4.3. Sampling and Measurements of Plasma TMAO and SCFA Levels

Initial fasting peripheral blood samples were collected from all participants within 24 h after the onset of AIS. Plasma TMAO and SCFA levels were measured through liquid chromatography with triple-quadrupole mass spectrometry (LC-MS). In summary, for all detection and quantification of analytes, Waters ACQUITY UPLC system (Waters Corporation, Milford, MA, USA) coupled with a tandem MS (Finnigan TSQ Quantum Ultra triple-quadrupole MS, Thermo Electron, San Jose, CA, USA) in combination with the Xcalibur software (Thermo Xcalibur version 2.2, ThermoFinnigan, Bellefonte, PA, USA) was used. The details of the experimental process, calibration procedures, quality control measurems, and limits of quantification were provided in Appendix A for TMAO; Appendix A for SCFAs). Eight types of SCFAs were measured: formate, acetate, propionate, isobutyrate, butyrate, 2-methylbutyrate, isovalerate, and valerate.

### 4.4. Statistical Analyses

Patient demographics, medical history, and blood biochemistry were categorized into categorical and continuous variables. These data were compared between patients with mild-to-moderate disability (mRS score = 0–3) and severe disability (mRS score = 4–6) by using the two-tailed *t*-test for continuous variables and the chi-squared test for categorical variables. The NIHSS score, type of recanalization therapy, time from symptom onset to treatment, good angiographic reperfusion (mTICI 2b-3), and site of arterial occlusion were compared between the groups.

To investigate the link between TMAO, SCFAs, and stroke severity, the Spearman rank correlation was employed to establish correlations between TMAO and SCFA levels and the NIHSS score. In addition, TMAO and SCFA levels were compared between mild-to-moderate and severe disability groups by using a two-tailed *t*-test. A multivariate logistic regression model was conducted to identify the independent association between the isovalerate level and the outcome of severe disability. Multivariable-adjusted OR and 95% CI for the association between isovalerate level and severe disability were reported in three models. In model 1, we adjusted for age, sex, and BMI. In model 2, we further adjusted for other vascular risk factors. In model 3, the treatment of EVT, including EVT only and combined IVT and EVT, was added to the regression model. Furthermore, the chi-squared test or Fisher’s exact test with odds ratio analysis was performed to assess the prognostic value of isovalerate for functional outcomes in the total patient population and in subgroups. Subgroup analyses were stratified by variables, including age (<75 and ≥75), sex, BMI (<27 and ≥27), comorbid vascular risk factors, initial NIHSS (<8 and ≥8), and types of recanalization therapy. The difference between subgroups was calculated by adding a subgroup by isovalerate interaction term to the regression model. Significance was set at a two-sided value of *p* < 0.05.

### 4.5. Ethical Approval

This study was approved by the Institutional Review Board of Kaohsiung Medical University Hospital (KMUHIRB-E(I)-20200424) and was conducted in accordance with the Declaration of Helsinki for experiments involving humans. Written informed consent was obtained from patients or their legal representatives before enrolment.

## 5. Conclusions

Our results indicate an association among the plasma isovalerate level, acute AIS severity, and early functional outcomes after recanalization therapy for AIS. An elevated plasma isovalerate level was correlated with mild stroke severity and disability after recanalization therapy for AIS. However, the pathophysiology that occurs with isovalerate immediately after AIS remains to be fully understood. The change in the gut microbiota corresponding to the isovalerate level and the effect of recanalization therapy for AIS warrant further analysis.

## Figures and Tables

**Figure 3 ijms-24-10796-f003:**
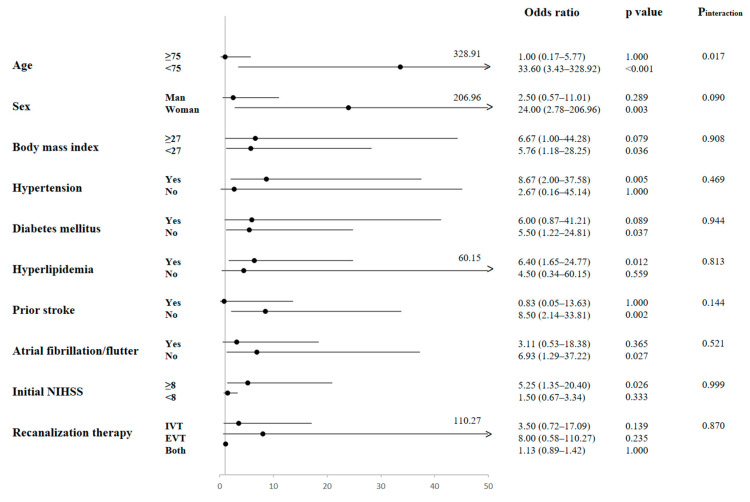
Odds ratio of a plasma level of isovalerate > 1.54 μM to mild-to-moderate disability in subgroups of enrolled patients. Odds ratios with 95% confidence intervals and *p* values are presented. NIHSS, National Institute of Health Stroke Scale, EVT, endovascular thrombectomy; IVT, intravenous thrombolysis.

**Table 1 ijms-24-10796-t001:** Baseline demographic characteristics, prevalence of vascular risk factors, and stroke severity of study participants stratified by functional outcomes.

Characteristics	Total	Mild-to-Moderate Disability (n = 23)	Severe Disability (n = 33)	*p* Value
Age, years, mean (±SD)	70.7 ± 13.0	69.0 ± 12.7	71.9 ± 13.3	0.410
Sex, male, n (%)	32 (57.1%)	13 (56.5%)	19 (57.6%)	0.938
BMI, mean (±SD)	25.9 ± 3.6	26.3 ± 3.6	25.6 ± 3.6	0.501
Prior vascular risk factors				
Hypertension, n (%)	46 (82.1%)	16 (69.6%)	30 (90.9%)	0.040 *
Diabetes mellitus, n (%)	23 (41.1%)	8 (34.8%)	15 (45.5%)	0.425
Hyperlipidemia, n (%)	42 (75.0%)	19 (82.6%)	23 (69.7%)	0.272
Prior stroke, n (%)	11 (19.6%)	3 (13.0%)	8 (24.2%)	0.299
Atrial fibrillation/flutter, n (%)	27 (48.2%)	7 (30.4%)	20 (60.6%)	0.026 *
Smoking, n (%)	10 (17.9%)	2 (8.7%)	8 (24.2%)	0.135
Initial NIHSS, median (IQR)	15 (10.3–21)	10 (6–15)	18 (14–22.5)	0.002 **
Discharge NIHSS, median (IQR)	7 (3–15)	2 (1–4)	13 (8.8–25)	<0.001 **
Recanalization therapy, n (%)				0.006 **
IVT only	30 (53.6%)	18 (78.3%)	12 (36.4%)	
EVT only	15 (26.8%)	4 (17.4%)	11 (33.3%)	
Both IVT and EVT	11 (19.6%)	1 (4.3%)	10 (30.3%)	

Abbreviations: BMI, body mass index; EVT, endovascular thrombectomy; IQR, interquartile range; IVT, intravenous thrombolysis; NIHSS, National Institute of Health Stroke Scale; SD, standard deviation. * indicated *p* < 0.05. ** indicated *p* < 0.01.

**Table 4 ijms-24-10796-t004:** Multivariate logistic regression analyses for determining severe disability.

	Model 1			Model 2			Model 3		
	OR	95% CI	*p* Value	OR	95% CI	*p* Value	OR	95% CI	*p* Value
Isovalerate	0.32	0.13–0.79	0.013 *	0.27	0.09–0.79	0.017 *	0.32	0.11–0.95	0.041 *
Age	1.02	0.97–1.07	0.565	1.02	0.96–1.08	0.590	1.05	0.98–1.12	0.196
Sex, male vs. female	1.80	0.51–6.45	0.364	1.43	0.35–5.85	0.620	2.38	0.47–12.09	0.296
Body mass index	0.92	0.77–1.11	0.378	0.84	0.68–1.04	0.113	0.86	0.68–1.09	0.213
Hypertension				3.29	0.52–20.98	0.208	2.92	0.42–20.29	0.279
Diabetes mellitus				2.71	0.62–11.90	0.188	2.66	0.48–14.64	0.260
Hyperlipidemia				0.38	0.06–2.37	0.297	0.56	0.07–4.34	0.577
Smoking				6.22	0.71–53.95	0.097	11.85	0.94–148.83	0.056
Prior stroke				1.21	0.21–7.01	0.831	1.16	0.15–8.71	0.889
EVT, vs. IVT only							10.19	1.71–60.78	0.011 *

Model 1: adjusted for demographic data, namely age, sex, and body mass index. Model 2: adjusted for the covariates in model 1, comorbid vascular risk factors, smoking status and prior stroke. Model 3: adjusted for the covariates in model 2 and the treatment of endovascular thrombectomy (endovascular thrombectomy only and combined intravenous thrombolysis and endovascular thrombectomy). Abbreviations: CI, confidence interval; EVT, endovascular thrombectomy; IVT, intravenous thrombolysis; OR, odds ratio. * indicated *p* < 0.05.

## Data Availability

The data presented in this study are available on request from the corresponding author. The data are not publicly available due to privacy.

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
