# Peer review of "The Prognostic Biomarkers of Plasma Trimethylamine N-Oxide and Short-Chain Fatty Acids for Recanalization Therapy in Acute Ischemic Stroke"

_ijms, 2023, doi:10.3390/ijms241310796_

Round 1

Reviewer 1 Report

The work is devoted to the search for a possible correlation between the levels of trimethylamine N-oxide (TMAO) and short-chain fatty acids (SCFAs) and the severity of the consequences of a stroke. These substances are metabolites of a number of microorganisms that inhabit the intestines. Indeed, in recent years, the attention of many researchers has been riveted to the study of the microbiota-gut-brain axis. Medical researchers see intelligent and conscious intervention in its functioning as an opportunity for a more gentle treatment of patients with serious illnesses such as stroke. The authors of this work were able to show for the first time that elevated plasma isovalerate level was correlated with mild stroke severity and disability after recanalization therapy for AIS. However, to bring it into the practice of therapy, to use it in any useful way for patients, time still has to pass and additional experiments must be performed. In general, this is a well-executed study with adequately selected and concisely described methods. It deserves the attention of the scientific community, and may be useful in terms of developing treatment protocols for stroke patients, taking into account the connection along the microbiota-gut-brain axis.

Reviewer 2 Report

In the manuscript by Chou et al., " The Prognostic Biomarkers of Plasma Trimethylamine N‐oxide and Short-Chain Fatty Acids for Recanalization Therapy in Acute Ischemic Stroke", the authors have evaluated the role of TMAO and isovalerate (SCFA) post recanalization in AIS. Overall, it is a good study along with relevant background and discussion. Although role of SCFA is known is stoke, it is novel findings in stroke with post-recanalization. However, I have concerns with the following points.

1.       Table 1: Please provide the data for female as it is only for male.

2.       Authors have used Man, woman, male, and female. It is better to write male and female in the manuscript.

3.       Although authors have discussed, please discuss in detail of Figure 3 findings.

4.       Please refer recent publication  (e.g  doi: 10.1155/2023/6297653).

Reviewer 3 Report

This is an interesting study to determine whether gut microbiota metabolites are associated with short-term outcomes in patients undergoing reperfusion therapy.

Here are some comments for the authors to improve the quality of the manuscript.

1. While previous studies have shown higher TMAO levels in patients with more severe stroke, this study did not find a similar association. The authors attribute this difference to the different TMAO levels among studies and to participants with severe stroke who underwent recanalization therapy compared to previous studies.

However, given previous findings, it is unexpected that TMAO levels in this study, including more severe stroke patients, were lower than in previous studies and even lower than in the healthy control group in previous studies. Therefore, I suggest describing more details about the TMAO measurement methods and the timing of sample collection after stroke onset in the Methods section. In addition, an additional explanation for the discrepancy in TMAO levels between this study and previous studies is needed.

2. The study described that an increased isovalerate level, based on the median isovalerate cut-off, was associated with an increase in mild to moderate short-term disability (lines 181-182).

It is assumed that logistic regression analysis was performed to obtain these results. If other confounding variables were adjusted for in the multivariable analysis, it is crucial to mention which variables were corrected for. In particular, it may be necessary to adjust for the type of recanalization therapy and successful recanalization.

If the reported association between isovalerate level and functional status is based solely on the results of univariate analysis, it would be difficult to conclude that isovalerate level is indeed related to functional status without considering other variables.

3. Result section 2.1 states that a total of 56 individuals received recanalization therapy. However, in Result 2.2, it is described that 36 received IVT, 15 received EVT, and 11 received a combination of IVT and EVT. These numbers do not match. Please check the numbers.

4. I recommend including the recanalization therapy types (IVT, EVT, IVT+EVT) in Table 1.

5. In subgroup analysis, isovalerate seems to show no directional difference in effect between groups except in the subgroup of prior stroke. The p-value for the interaction should be shown in Figure 3.

Please consider consulting a statistician.

6. In the method section, please be specific about the TMAO, SCFA measurement method.

It is required to mention the specific equipment, instrument models, and reagents used in the measurement process. Additionally, any quality control measures, calibration procedures, and limits of detection or quantification should be provided to ensure the reliability and accuracy of the TMAO/SCFA measurements.

Minor editing of English language required.

Round 2

Reviewer 3 Report

Thank you for the authors' efforts.

The manuscript has been appropriately revised and appears worthy of publication.